# Does Musicality Assist Foreign Language Learning? Perception and Production of Thai Vowels, Consonants and Lexical Tones by Musicians and Non-Musicians

**DOI:** 10.3390/brainsci13050810

**Published:** 2023-05-16

**Authors:** Antonia Götz, Liquan Liu, Barbara Nash, Denis Burnham

**Affiliations:** 1MARCS Institute for Brain, Behaviour, and Development, Western Sydney University, Penrith, NSW 2751, Australia; a.goetz@westernsydney.edu.au (A.G.); l.liu@westernsydney.edu.au (L.L.); barbara.nash@ukbb.ch (B.N.); 2School of Psychology, Western Sydney University, Penrith, NSW 2751, Australia; 3Center for Multilingualism in Society across the Lifespan, University of Oslo, 0313 Oslo, Norway; 4Centre of Excellence for the Dynamics of Language, Australia Research Council, Majura Park, ACT 2609, Australia; 5University Children’s Hospital Basel (UKBB), 4056 Basel, Switzerland

**Keywords:** formal music training, language learning, perception and production, consonants, vowels and lexical tones

## Abstract

The music and spoken language domains share acoustic properties such as fundamental frequency (f0, perceived as pitch), duration, resonance frequencies, and intensity. In speech, the acoustic properties form an essential part in differentiating between consonants, vowels, and lexical tones. This study investigated whether there is any advantage of musicality in the perception and production of Thai speech sounds. Two groups of English-speaking adults—one comprising formally trained musicians and the other non-musicians—were tested for their perception and production of Thai consonants, vowels, and tones. For both groups, the perception and production accuracy scores were higher for vowels than consonants and tones, and in production, there was also better accuracy for tones than consonants. Between the groups, musicians (defined as having more than five years of formal musical training) outperformed non-musicians (defined as having less than two years of formal musical training) in both the perception and production of all three sound types. Additional experiential factors that positively influenced the accuracy rates were the current hours of practice per week and those with some indication of an augmentation due to musical aptitude, but only in perception. These results suggest that music training, defined as formal training for more than five years, and musical training, expressed in hours of weekly practice, facilitate the perception and production of non-native speech sounds.

## 1. Introduction

The domains of music and speech are highly intertwined. Given such interweaving, it is reasonable to expect that there could be transfer between the two domains, and there is empirical evidence to support this. For example, rhythmic priming has been shown to facilitate grammatical processing in children with and without developmental language disorder [1]. In addition, musical training in eight- to ten-year-old children has been shown to have a positive effect on speech segmentation abilities for pseudo-words with a unique melodic contour [2]. Furthermore, after twelve months of musical training, eight- to ten-year-old children show improved neural processing in the perception of durational differences in consonants and in vowels [3], and after six months of musical training, eight-year-old children showed enhanced reading and pitch-discrimination abilities in speech alongside enhanced neural responses to subtle pitch variations in speech [4,5]. Similarly, musically trained ten-to-eleven-year-olds exhibited larger event-related potentials to musical harmonic violations, as well as linguistic grammatical violations, compared to their non-musically trained counterparts [6]. Reciprocally, the speaker’s familiarity with the features of their native language has been suggested to influence their pitch perception in music [7,8] and to influence rhythmic [9,10] and melodic [11] musical production.

In this study, we investigate the cross-domain relationships between speech and music via a comparison of musicians versus non-musicians’ perception and production of multiple speech contrasts, namely Thai (i) consonants, (ii) vowels, and (iii) lexical tones. In the following, we first discuss the results of previous studies comparing musicians and non-musicians regarding speech perception or production. In the review, we first address perception, then production.

Previous studies found divergent results between musicians and non-musicians in the perception of segmental features such as consonants and vowels. For example, while Marie and colleagues [12] showed a positive relationship between musicality and consonant and vowel discrimination, Delogu et al. [13,14] showed that musicians have an advantage over non-musicians in discriminating pitch differences, but no advantage in discriminating segmental contrasts. It is interesting to note that the differences between consonants and vowels may have been a result of the underlying nature of these speech sounds. Vowels appear to be perceived less categorically than consonants [15,16,17]. However, the divergent results between musicians and non-musicians’ ability to perceive differences in segmental contrasts diminishes when the contrasts mainly involve temporal information (e.g., duration), with musicians showing better discrimination than non-musicians (e.g., [18,19]). For example, in a cross-linguistic study, Sadakata and Sekiyama [19] compared musicians and non-musicians who were native speakers of Japanese or of Dutch on their discrimination of Japanese length differences in nasals (e.g., konyaku—konnyaku) and plosives (hakaku—hakkaku), as well as quality differences in Dutch vowels (put—pet, tut—toet). They found that musicians, regardless of their native language, readily distinguished differences in the length of plosives better than non-musicians, but the musicality advantage for vowel quality differences was much less robust. As temporal information is an essential component in both the musical and the linguistic domains, these results may indicate that there is transfer between the two domains. Moreover, it appears that such transfer effects may be specific to the acoustic properties that play an important role in both domains. The transfer effect for pitch and durational acoustic properties may not be surprising given that the main music notation system in Western tonality is tailored to encode pitch and durational information [20]. Similarly, aural-skill training and assessment in music education is predominantly focused on pitch and rhythmic skills [21].

Further support for musicians’ superior discrimination being based on temporal information is evident in studies dealing with voice onset time (VOT). VOT is the temporal delay between the release of closure at the place of articulation and the onset of voicing [22]. In many languages, VOT differentiates voiced from voiceless plosives. The distinctions are language-specific, such that some languages (such as Thai) have a three-way phonemic distinction between voiced, voiceless, and voiceless aspirated, e.g., /b/-/p/-/p^h^/, sounds, while other languages (such as English) have only a two-way phonemic distinction between voiced and voiceless aspirated, /b/-/p^h^/ sounds. For example, Dittinger et al. [18] compared French musicians and French non-musicians (for whom only the voiced-voiceless, e.g., /b/-/p/, distinction is phonemic) on a non-native aspiration contrast, /p/-/p^h^/, and found that the musicians showed better behavioural discrimination, as well as stronger neural responses, than the non-musicians.

While all the world’s languages use segments to specify meaning, suprasegmental variations, such as tone and pitch accent, are not used universally at the lexical level. Nevertheless, tone languages comprise 60–70% of the world’s languages [23] and are spoken by more than 50% of the world’s population [24]. Despite their similar lexical functions, the determinants of variation in consonants and vowels and in tones differ. The main acoustic dimensions on which consonants and vowels differ are spectral and durational, whereas tone variation is predominantly based on pitch—f0 height and f0 movement over time (contour) (see [25]). As tones are mainly carried on vocalic elements of a speech signal, it is hypothesised that they are similar to vowels and are perceived less categorically. [26,27]. Given these similarities, musicality could potentially provide a more nuanced understanding of the underlying distinctions between consonants, vowels, and tones. As pitch is also an important feature of music, especially with respect to melody, it might be expected that people with musical training, and perhaps those with greater musical aptitude, should be better able to transfer their skill in the musical domain to the perception of tones than people without such musical training or aptitude.

Indeed, previous studies have shown that musically trained participants are better at perceiving linguistic pitch than those without musical training (e.g., [28,29]). For example, Burnham et al. [29] compared the discrimination of Thai tones by three groups of non-tone language listeners: those with no musical training, those with musical training without absolute pitch perception, and those with musical training with absolute pitch perception. Both groups of trained musicians performed better than the non-musicians, and the musicians with absolute pitch showed an additional advantage. Similar results have been found at the neurophysiological level. Chandrasekaran et al. [30] found that both tone language (Mandarin) non-musicians and non-tone language musicians showed greater neural responses to Mandarin within- and between-category tone contrasts than non-tone language non-musicians (see also [31]).

To summarise, musicians have an advantage over non-musicians in speech perception, mainly when contrasts are based on the overlapping acoustic properties of speech and music: fundamental frequency and duration. Such an overlap presumably facilitates the perceptual transfer between the music and speech domains.

In addition to speech perception, possible transfer effects have been investigated in speech production, albeit to a much lesser extent. Gottfried [32] compared the production of Mandarin tones between musicians and non-musicians and found that musicians’ productions were classified by Mandarin-language raters, blind to the speakers’ musical status, as more native-like than those of non-musicians. Nikjeh et al. [33] showed that instrumental and vocally trained musicians outperformed non-musicians in the production of pitch. With respect to consonant and vowel production, memory for rhythm and amount of musical practice were also found to be predictors of the quality of pronunciation of non-native speech sounds [34,35,36]. For example, Jekiel and Malarski [34] found that after a period of training, the accuracy of the production of non-native vowels by Polish speakers of L2 English was related to their level of formal music training. In non-native consonants, Slevc and Miyake [36] investigated Japanese learners of English perception and production of the English phonemes /r/ and /l/, along with their musical ability, as indexed by their perception of and memory for tones and chords. They found that the Japanese learners of English could indeed perceive and produce /r/ and /l/, and that musical ability is a significant predictor of the perception and production of this non-native contrast.

Most existing studies have compared the perception and/or production performance of musically trained individuals and that of individuals without such training. However, the range of musicality is very broad and cannot be distinguished solely on the dimension of formal musical training; individual differences among various dimensions play a major role in perception and production performance in language learning (e.g., [37]). Similarly, it is suggested that the type of musical training and experience influences pitch perception and production abilities, with individuals who are vocally trained showing greater accuracy in pitch production than musicians who are not vocally trained [33,38]. Individual differences can be assessed more comprehensibly if, in addition to musical training, the influence of musical aptitude is also taken into account. Musical aptitude has been shown in the perception and production of non-native speech sounds [35,36,39,40,41,42]. For example, Milovanov et al. [35] showed that the musical aptitude of Finnish-speaking participants is closely related to their foreign language proficiency, especially in the production of non-native English consonants and vowels.

The aim of this study is to investigate whether musicality, in terms of both formal music training and musical aptitude, influences both the perception and production of non-native speech sounds, specifically Thai consonants, vowels, and tones. To this end, 36 English-speaking adults—one group of formally trained musicians and another of non-musicians—are assessed for their musical aptitude and tested for their perception and production of various Thai consonants ([b], [p], and [p^h^]), vowels ([u:], [i:], and [ɔ:]), and tones (low level, high level, and mid-level tone). As both speech and music abilities involve perception *and* production, we investigate both of these here, first in a perception task, then in a production task. Speech perception was evaluated in an identification task that enabled the assessment of not only their ability to discriminate between two speech sounds, but also their ability to accurately identify the speech sounds. This task is of particular relevance in evaluating language learning as the ability to correctly identify speech sounds is critical for the accurate production of the corresponding speech sounds. The goal of the study is to determine whether, and to what extent, musicality (formal training and/or aptitude) predicts the (a) perception and/or (b) production in (i) consonants, (ii) vowels, and/or (iii) tones.

## 2. Method

### 2.1. Participants

A total of 36 native Australian English language participants were tested; 18 musicians (10 female, 8 male, average age: 27.8 years) and 18 non-musicians (10 female and 8 male, average age: 24.6 years). Participants gave their written informed consent regarding all stages of the experiment (data collection, analysis, publication), and the experiment was covered by Ethics Approval from Western Sydney University (HREC 06/65). All participants were students at the university and received course credit for their participation. None had previous exposure to a tone language. Musicians were defined as instrumentalists/singers having at least five years of continuous formal musical training with traditional instruments, e.g., no electronic instruments (*M* = 15.7 years, *sd* = 10.62). Non-musicians were defined as having no more than two years of musical training (*M* = 0.11 years, *sd* = 0.47). None had any self-reported hearing or speech/language problems. Participants were tested individually in a single session, in a sound-attenuated testing cubicle in laboratories at the Western Sydney University. Participants were given a questionnaire, which included demographic and sensory information and musical history, and tests of musical aptitude and musical memory. They were then tested for their perception and production of tones, consonants, and vowels. The order of these tests was counterbalanced between participants. Testing took a total of 75 min. Stimuli for all tasks were presented on a laptop computer (Compaq Evo N1000c) over headphones (KOSS UR20) at a self-adjustable listening level. Details of each test are given below.

### 2.2. Stimuli

A total of 27 Thai syllables were recorded, each consisting of a consonant-vowel syllable with an accompanying tone (CV + T). Three levels of each of the three features were included, giving rise to 3 × 3 × 3 = 27 combinations. As shown in Table 1, the variations were, for vowels, three vowel-qualities ([u:], [i:], and [ɔ:]); for consonants, three levels of voicing—([b], [p], and [p^h^]); and for tones, three levels (low level, high level, and mid-level tone).

One male (23 years) and one female (22 years), both native speakers of Central (Bangkok) Thai, recorded multiple tokens of the 27 syllables. From these, a native Thai speaker with phonetics training selected tokens best matched in terms of f0 contour and duration to be used in the experiment. In both the perception and the production tasks, the male participants were presented with the male speaker stimuli and the females were presented with the female speaker stimuli so that the f0 of the participants’ productions could be more directly compared to the native speaker models.

### 2.3. Speech Perception Procedure

Perception was measured in an identification task presented in the DMDX [43] experimental environment. Participants were asked to press the LEFT SHIFT key for one kind of sound, the SPACEBAR key for another kind of sound, and the RIGHT SHIFT key for a third kind of sound. The LEFT, SPACEBAR, and RIGHT keys were labelled, for the consonant task, “ph” (for the voiceless aspirated bilabial), “p” (voiceless unaspirated bilabial stop), and “b” (prevoiced bilabial stop); for the vowel task “i”, “o”, and “u”, and for the tone task with the tone contours, in stylised form.

There were three parts: consonant identification, vowel identification, and tone identification, each consisting of a practice block, a training block, and a test block. In the practice block, nine items were presented, with three of each of the contrasting sounds in the relevant dimension. In the training block, a criterion of three consecutive correct responses was required before testing continued. The same sounds were presented as in the practice block, and feedback was provided. In the test block, no feedback was given, there were two repetitions of each of the 27 items presented in random order, and responses timed out after 7000 ms, without replacement. Given the 27 different stimuli, two repetitions, and three parts of the test, participants were required to identify 162 items in total. The order of parts (tone, consonant, vowel) was counterbalanced between subjects. Each of the tasks took between 4 and 5 min, depending on the participants’ pace.

### 2.4. Speech Production Procedure

In the production task, the same 27 stimuli were used as in the perception task, again with male speaker stimuli for males and female speaker stimuli for females. The stimuli to be imitated were presented in Power Point. First, all 27 stimuli were presented in random order and participants were required to repeat each sound. Then, the sounds were presented systematically. First, the different consonant sounds were introduced in succession with a crossed-out microphone presented on the display, and the participant was instructed to listen and not repeat the sounds. Then, the same three sounds were presented separately, and the listener was required to repeat each of the sounds. Similarly, in the vowel phase, the three stimuli, varying in vowel colour, were presented, and in the tone phase, the three stimuli differing in lexical tone were presented for listening then repeating. After the separate presentations of consonants, vowel, and tones, all sounds were presented again in random order. Thus, each participant was required to produce five productions of each of the 27 syllables, comprising a total of 135 productions. The production task took around 12 min.

Two native Thai phoneticians were employed to rate each participant’s productions for each syllable on a scale of 1 (very bad) to 5 (very good). Reliability between raters was high (r = 0.83). The scores for each of the five productions for each participant were averaged, resulting in a mean score between 1 and 5 for each participant for each of the 27 sounds.

## 3. Results

### 3.1. Perception

In the speech perception task, the listeners were required to identify consonants ([b], [p], [p^h^]), vowels ([i:], [ɔ:], [u:]), and tones (low, mid, high). For the analyses, we used the lme4 package ([44], version 1.1-26) in R ([45] version 4.0.4). General Linear Mixed Effects regression models were constructed with the maximal random and fixed factor structure, with accuracy as the dependent variable. The model contained the fixed effect of musicality (musicians coded as −0.5 and non-musicians as +0.5); for the speech sound type, with its three levels, we compared tones and consonants against vowels (tones and consonants coded as −1/3, and vowels as 2/3); and tones versus consonants (tones coded as −0.5, and consonants as 0.5 and vowels as 0). Coding was performed using the general inverse [46]. In addition, we entered the tone perception scores, rhythm scores (both reflecting the musical aptitude), years of musical training, and weekly hours of musical training (both reflecting formal music training) as scaled continuous variables. Participant was entered as a random effect, with sound type as random slope. For an overall perception, the estimates for the fixed effects were extracted from the model (see Table 2).

The percent of correct identifications of consonants, vowels, and tones by the musicians and non-musicians are shown in Figure 1. The analyses showed that musicians performed significantly better than non-musicians (*β* = 0.069, *z* = 3.248, *p* = 0.001). In addition, the participants were better at identifying vowels than tones and consonants (*β* = 0.213, *z* = 10.919, *p* < 0.001), but there was no difference in the identification performance between tones and consonants (*β* = 0.017, *z* = 0.853, *p* = 0.399). There were no significant interactions of musicians’ superiority with speech type, so it can be concluded that the musicians’ superiority extended across all speech types, and the speech type differences extended across both groups. With respect to formal music training versus aptitude, hours of weekly musical training had a significant positive effect on speech sound identification (*β* = −0.036, *z* = −2.154, *p* = 0.032), reinforcing the notion that the effect of musicality is due to formal music training, not aptitude. However, it must be noted that one of the two measures of musical aptitude—the rhythm but not tone score—was marginally significant (*p* = 0.051). No other predictors, nor their interactions, were significant.

### 3.2. Production

The rated speech production performance of vowels, consonants, and tones were analysed with the same factors and with the same contrast coding as for the speech perception analysis. The model output can be seen in Table 3 and in Figure 2.

Overall, the musicians produced the speech sounds more accurately than the non-musicians (*β* = 0.257, *z* = 4.123, *p* < 0.001). In addition, across both groups, the participants produced vowels more accurately than consonants and tones (*β* = −0.068, *z* = −2.068, *p* < 0.001), and in turn, tones more accurately than consonants (*β* = 0.060, *z* = 2.771, *p* < 0.01). There were no significant interactions of these speech type contrasts with musicianship, so the musicians’ superiority extended across all speech types and speech type differences across both groups. Finally, as in the perception analyses, hours of musical training had a positive impact on the production scores (*β* = 0.131, *z* = 2.665, *p* = 0.011), irrespective of musicianship. However, in contrast to the perception results, neither of the two aptitude scores, rhythm nor tone scores, were, nor did they approach, significance.

### 3.3. Linking Perception with Production

In order to explore the link between perception and production accuracy, we ran Pearson correlation tests. The results revealed a significant positive correlation between perception and production accuracy (*r* (34) = 0.121, *p* = 0.03) across the groups, suggesting a facilitative link between the perception and production of consonants, vowels, and tones.

## 4. Discussion

In this study, we investigated musicians versus non-musicians’ perceptual identification and speech production of three speech types: consonants, vowels, and tones. In addition, the impact of musical aptitude and musical memory and the duration of musical training and weekly exposure to music on perception and production were evaluated. Although the perception and production performances were analysed separately, here they are discussed together, in three sections, in terms of the separate and independent effects of (i) Musicianship and (ii) Speech Type, and how each of these relate to the specificity of any (iii) Cross-Domain Transfer. These three issues are discussed in turn.

### 4.1. Musicianship

Musicians (instrumentalists/singers with at least five years of continuous formal musical training) performed significantly better than non-musicians (no more than two years of musical training) in both perception and production. As there were no significant interactions of musician/non-musician with the perception or production of consonants, vowels, or tones, it can be concluded that English-language adults show a robust musician versus non-musician advantage for both the perception and production of all speech types that signal lexical meaning in Thai-segments (consonants and vowels) and supra-segmentals (tones). The absence of an interaction between Thai sound contrasts and musicality could be attributed to the fact that, among musicians, pitch perception and production abilities differ depending on the type of musicality, with vocalists showing greater accuracy in pitch perception (particularly vocal pitch matching) and production than non-vocalists [31,36]. In our study, we did not systematically differentiate between musicians with vocal versus instrumental training, and a more refined categorisation of different musicality types could possibly reveal that musicians with vocal training excel in lexical tone perception and production, while still performing similarly to instrumentalists for consonants and vowels. However, in our sample, ten of the 18 musicians had both vocal and instrumental training, while only eight had purely instrumental training, thus precluding a post hoc examination of vocal and instrumental training. Future investigations of speech sound discrimination and production could investigate this issue by selecting musical participants who have only vocal training versus only instrumental training, while possibly including another group with training in both.

The overall musical advantage here is in line with the results showing a positive effect of formal music training on non-native vowel production [34], pitch perception, and production [33]. Moreover, the musicality advantage here appears to be based mainly on formal music training because, of all the additional variables measured here, only hours of training per week was found to exert a positive effect on performance, and this was the case in both perception and production for both musicians and non-musicians. There is also some indication that this effect is one of current musical activity, as total years of training was not significant for either perception (*p* = 0.623) or production (*p* < 0.084). However, it must be noted that one aspect of musical aptitude (rhythm scores) had a marginally significant effect on the perception (*p* = 0.051), but not production, performance, so it is possible that some degree of musical aptitude, independent of experience, contributes to the musicality advantage in the perception of Thai consonants, vowels, and tones.

### 4.2. Speech Type

Across the participants—including both musicians and non-musicians—there was a hierarchical structure in the relative ease of the perception and production of consonants, vowels, and tones. In terms of perception, the Thai vowels were more accurately identified than the consonants and tones. In terms of production, vowels were also produced with better accuracy than consonants and tones; in addition, tones were produced more accurately than consonants. Thus, both the perception and production were most accurate for vowels, least accurate for consonants, with tones being intermediate in perception and equivalent to vowels in production.

There are two possible explanations for the outstanding performance in the vowel identification and production results. The first derives from an acoustic/articulatory perspective, irrespective of native language effects: the vowels /i/ and /u/ form extreme points in the vowel space by differing maximally in their tongue positions and lip roundedness (/i/ being a front and unrounded vowel, whereas /u/ is a back and rounded vowel). The third vowel (/ɔ/) differs from the other two vowels in terms of its openness and thus has higher first formant frequencies compared to both closed vowels. These articulatory characteristics may lead to considerable differences in first and second formants, resulting in large acoustic salience differences between the vowels and thus making it easier to perceive and identify those vowels. In contrast, the acoustic distance of consonants is relatively smaller. The Thai consonants used in the present study differ in terms of their initial voicing: /b/ is characterised as a prevoiced plosive, /p/ as a voiceless plosive, and /p^h^/ as a voiceless but aspirated plosive. Typically, the acoustic parameter for differentiating between these consonants is the VOT, which ranges in the used exemplars along an acoustic continuum, which may have made it difficult for the participants to identify the consonants correctly. The intermediate role of tones can also be considered in relation to the acoustics. As for consonants, the tones move along an f0 spectrum between low, mid, and high. However, the acoustic distance of these sounds may be relatively greater than for consonants and smaller than for vowels, resulting in an intermediate position in perceptual and production ability.

The second explanation has a language-specific basis. Despite the fact that all of the sounds are produced by Thai native speakers, the vowels, in particular, have close equivalents in the English language and thus may be assimilated to native vowels categories [47,48]. Non-native vowels that are assimilated to native vowel categories can be identified more easily because listeners have already formed phoneme categories for these sounds. Similarly, tone contrasts may have been assimilated to native intonation contrasts [49], whereas the three-way consonant contrast is non-phonological and may have led to more difficulties regarding which sounds might be similar to their native language categories.

### 4.3. Specificity of Cross-Domain Transfer of Musicality

In the perception tasks, we expected a superior identification of consonants involving temporal (VOT) differences by musicians on the basis of studies by Dittinger et al. [18]; and of tones on the basis studies by Alexander et al. [28] and Burnham et al. [29]. The evidence regarding vowels is less clear-cut; while a positive relationship between musicality and vowel discrimination has been found [12], Delogu et al. [13,14] showed that musicians better discriminate pitch differences than non-musicians in discriminating tones, but have no advantage in discriminating vowel differences. With respect to production, there is evidence that musicians’ production of non-native tones is better than non-musicians [32], and that the production of non-native vowels and consonants is related to a person’s memory for rhythm and amount of musical practice [34,36,40].

Our results show that, overall, musical training leads to a facilitation effect in a broad range of speech sounds. However, we did not find evidence of an interaction of specific speech types with musicianship. These results suggest that musicians excel in learning to perceive and produce foreign speech sounds as a product of a heightened auditory, possibly phonological, awareness, an awareness and facility that extends to the various types of sound distinctions that signal meaning in spoken language: consonants, vowels, and tones.

### 4.4. Further Research

In our results, there is a clear musician over non-musician music-to-speech transfer for all three types of speech sounds—consonants, vowels, and tones—in perception and production tasks in a particular first (English) to second (Thai) language learning situation. In addition, while there were differences in the performances across these speech types, these differences were independent of musicality. Further research is required to ascertain whether such an across-the-board advantage for musicians is specific to Thai, or to tone languages more generally, and whether it also occurs for non-tone languages. Additionally, given what appears to be a ceiling effect for the vowels in this study, further research (especially on the perception, but also the production) of more difficult vowels is required to tease out the nature of the differences between musicians and non-musicians’ facility with vowels. In order to explore the impact of musicality on speech sound categories, future investigations could consider utilising not only more complex vowels but also vowels that are not part of, or cannot be assimilated to, the participants’ native phonemic vowel system. Additionally, future studies could manipulate underlying cues, such as frequencies, timbre, or duration, to closely match those of the musical and speech domains.

Musicality can consist of learned experiential skills and/or more aptitude-based skills. Here, both were investigated. As there was a better performance for musicians (at least five years of continuous formal musical training) over non-musicians (no more than two years of musical training) in both perception and production, and as the performances in both were positively augmented by current training and practice (hours per week training), there is overarching evidence here for a major role of experience in music-to-speech transfer. However, it should be noted that, in the perception tasks, one aspect of musical aptitude (the rhythm score) was marginally significant (*p* = 0.051); therefore, further research is required to investigate the relative contributions of experience and aptitude in music-to-speech transfer, especially for perception tasks.

## 5. Conclusions

In terms of the perceptual identification and production of foreign (Thai) speech sounds by native English-language adults, there is a clear musicality advantage for the perception and production of Thai consonants, vowels, and tones. Here, ‘musicality’ consists predominately of formal music training in both perception and production, with some possible involvement of musical aptitude in the perception, but not production, of non-native speech sounds. This superior performance of musicians appears to be tied to a heightened awareness and facility in auditory tasks, for example, due to a greater habit of consciously reflecting on, memorising, mentally rehearsing, and reproducing sounds. This advantage extends to all three sound types that signal meaning in spoken language: consonants, vowels, and tones. Whether this heightened awareness and facility is specifically a heightened phonological awareness is yet to be determined.

## Figures and Tables

**Figure 1 brainsci-13-00810-f001:**
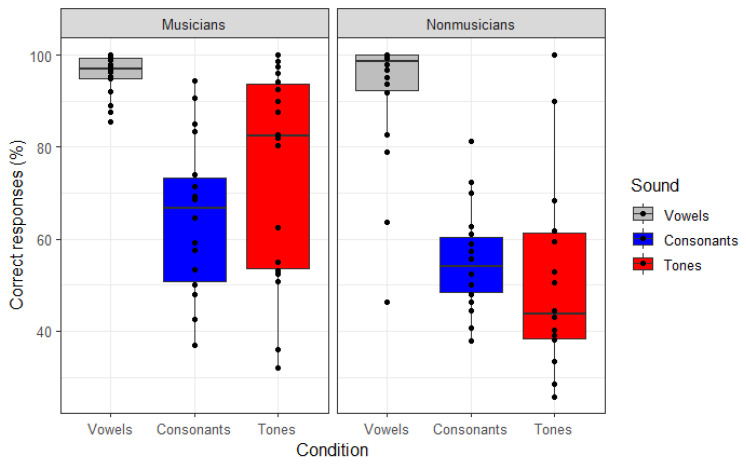
Identification accuracy across the three different speech sounds (vowels, consonants and tones) for musicians and non-musicians. Dots represents the mean response by an individual.

**Figure 2 brainsci-13-00810-f002:**
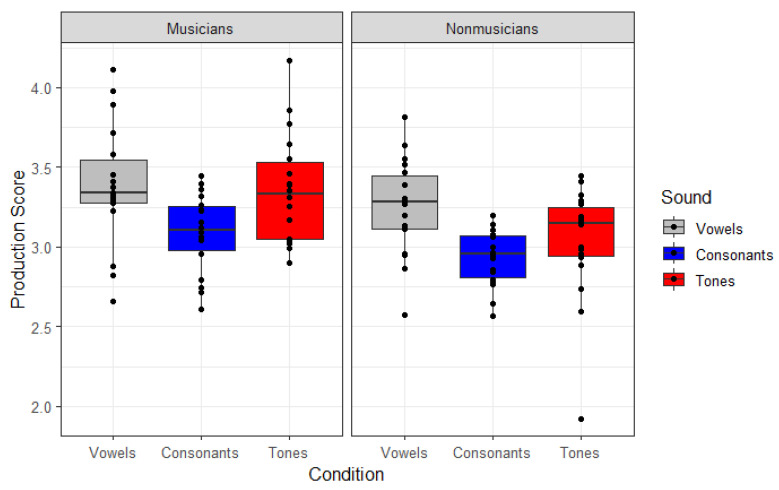
Rated production accuracy (1 to 5, five being the highest score) across the three different speech sound (Consonants, Vowels and Tones) for musicians and non-musicians. Dots represent the mean accuracy by an individual.

**Table 1 brainsci-13-00810-t001:** Matrix Showing Consonant, Vowel and Tone stimuli.

	Mid Tone (M)			Low Tone (L)			High Tone (H)		
	/i:/	/ɔ:/	/u:/	/i:/	/ɔ:/	/u:/	/i:/	/ɔ:/	/u:/
/b/	bi:0	bɔ:0	bu:0	bi:1	bɔ:1	bu:1	bi:3	bɔ:3	bu:3
/p/	pi:0	pɔ:0	pu:0	pi:1	pɔ:1	pu:1	pi:3	pɔ:3	pu:3
/p^h^/	phi:0	phɔ:0	phu:0	phi:1	phɔ:1	phu:1	phi:3	phɔ:3	phu:3

**Table 2 brainsci-13-00810-t002:** Parameters of the general linear mixed-effects regression for the perception of Consonants, Vowels, and Tones. For fixed effects, regression coefficients (*β*), their standard errors (SE), *z*-values and the respective *p*-values are given.

Predictor	Estimate	SE	df	*z*-Value	Pr (>|t|)
(Intercept)	0.723	0.012	280.670	58.265	<0.001
Musicians vs. Non-musicians	0.069	0.021	280.679	3.248	0.001
Tones and Consonants vs. Vowels	−0.213	0.020	64.681	−10.919	<0.001
Tones vs. Consonants	0.017	0.020	39.203	0.853	0.399
Tone score	0.019	0.022	280.655	0.886	0.377
Rhythm score	0.042	0.021	280.662	1.963	0.051
Years of Training	−0.010	0.019	280.663	−0.535	0.593
Hours per Week of training	−0.036	0.017	280.670	−2.154	0.032
Musicians vs. Non-musicians: Tones vs. Consonants	0.009	0.035	39.332	0.266	0.792
Musicians vs. Non-musicians: Tones and Consonants vs. Vowels	0.059	0.034	64.782	1.748	0.085
Tones vs. Consonants: Tone score	−0.010	0.035	39.004	−0.299	0.767
Tones and Consonants: Tone score	−0.040	0.034	64.525	−1.170	0.246
Tones vs. Consonants: Rhythm score	0.013	0.034	39.096	0.388	0.700
Tones and Consonants: Rhythm score	0.005	0.033	64.597	0.156	0.877
Tones vs. Consonants: Years Training	−0.040	0.031	39.109	−1.276	0.210
Tones and Consonants: Years Training	0.015	0.030	64.607	0.499	0.619
Tones vs. Consonants: Hours/Week	0.006	0.027	39.207	0.207	0.837
Tones and Consonants: Hours/Week	0.032	0.026	64.684	1.207	0.232

**Table 3 brainsci-13-00810-t003:** Parameters of the general linear mixed-effects regression for the production of Consonants Vowels, and Tones. For fixed effects, regression coefficients (*β*), their standard errors (SE), *z*-values and the respective *p*-values are given.

Predictor	Estimate	SE	df	*z*-Value	Pr (>|t|)
(Intercept)	3.173	0.036	36.366	87.726	<0.001
Musicians vs. Non-musicians	0.257	0.062	36.540	4.123	<0.001
Tones and Consonants vs. Vowels	−0.150	0.038	284.729	−3.985	<0.001
Tones vs. Consonants	0.090	0.033	284.647	2.771	0.006
Rhythm score	0.042	0.062	37.110	0.680	0.500
Years of Training	−0.102	0.058	39.408	−1.770	0.084
Hours per Week of training	0.131	0.049	37.223	2.665	0.011
Musicians vs. Non-musicians: Tones vs. Consonants	0.007	0.056	284.535	0.131	0.896
Musicians vs. Non-musicians: Tones and Consonants vs. Vowels	−0.064	0.065	284.809	−0.985	0.326
Tones vs. Consonants: Tone score	0.001	0.058	285.792	0.010	0.992
Tones and Consonants: Tone score	0.000	0.066	284.968	0.004	0.997
Tones vs. Consonants: Rhythm score	0.028	0.056	285.394	0.504	0.615
Tones and Consonants: Rhythm score	−0.025	0.065	284.887	−0.384	0.701
Tones vs. Consonants: Years Training	−0.061	0.051	284.502	−1.209	0.228
Tones and Consonants: Years Training	−0.118	0.063	286.018	−1.865	0.063
Tones vs. Consonants: Hours/Week	−0.004	0.044	284.337	−0.092	0.926
Tones and Consonants: Hours/Week	−0.007	0.052	286.009	−0.126	0.900

## Data Availability

The data presented in this study are available on request from the corresponding author.

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
