# Peer review of "Does Musicality Assist Foreign Language Learning? Perception and Production of Thai Vowels, Consonants and Lexical Tones by Musicians and Non-Musicians"

_brainsci, 2023, doi:10.3390/brainsci13050810_

Round 1
Reviewer 1 Report
I could not indicate particular critical points in this article. The article is enough interesting and the experiments and their purposes are clearly specified. The relationship between musical/rhythmic training and the ability to distinguish linguistic sounds does not seem to be unexpected, at least on the minimal basis that improving perceptual and attentional skills will influence linguistic tasks. Perhaps, a more in-depth investigation of the relationship between the structural properties of language and the structural properties of music could represent an important development of the research.
Author Response
Thank you very mich. We added more information into the limitation and future directions section.
Reviewer 2 Report
The authors provide an interesting study of the effects of musical training on speech perception and production for a foreign language. The study examines aspects of consonant, vowel, and tone perception and production for musicians and non-musicians and finds significant influences of musical training on consonant and tone perception but not vowel perception. The authors correctly note that the vowel contrasts, however, are similar to Australian English vowel categories and so sound type differences may not be a reliable finding. Overall, however, the study provides interesting new data for consideration.
Minor comments:
Page 2 discussion notes vowel versus consonant differences in findings on effects of musicality. It is probably worth noting that similar vowel versus consonant differences are known within categorical perception tasks in language regardless of musicality. However, similar non-categorical perception was reported for linguistic tones and so musicality might be an especially interesting place to examine the contrast between vowel, tone, and consonant perception.
References:
Abramson, A. (1979). Noncategorical perception of tone categories in Thai. In Journal of the Acoustical Society of America, 61, S66.
Pisoni, D.B. (1975). Auditory short-term memory and vowel perception. In Memory & Cognition, 3, 7-18.
Page 9 sections 4.1 and/or 4.3 the work of Dee Nikjeh is relevant to this discussion (and possibly to considerations in future work) as an effect of type of musical training and music experience influenced performance in production. (Also somewhat relevant in page 3 paragraph 4 discussing differences among musicians.)
Reference:
Nikjeh, D. A., Lister, J. J., & Frisch, S.A. (2009). The relationship between pitch discrimination and vocal production: Comparison of vocal and instrumental musicians. Journal of the Acoustical Society of America, 125, 328-338.
Page 9 section 4.4 on further research would benefit from noting that an additional study of Thai using the back unrounded vowel (which is not a vowel category in English) would provide a better test of whether sound category (vowel versus consonant versus tone) is differentially influenced by musical training.
Author Response
Thank you ver much. See our comments and modifications below, highlighted in blue.
Page 2 discussion notes vowel versus consonant differences in findings on effects of musicality. It is probably worth noting that similar vowel versus consonant differences are known within categorical perception tasks in language regardless of musicality. However, similar non-categorical perception was reported for linguistic tones and so musicality might be an especially interesting place to examine the contrast between vowel, tone, and consonant perception.
We included information about the divergent results on consonant and vowel perception and its relation to categorical perception on p.2
It is interesting to note that differences between consonants and vowels may have been a result of the underlying nature of these speech sounds. Vowels appear to be perceived less categorical than consonants [15]–[17]. However, the divergent results between musicians and non-musicians ability to perceive differences in segmental contrasts diminishes when the contrasts mainly involve temporal information (e.g., duration), musicians show better discrimination than non-musicians (e.g., [18], [19]).
(…) p.2-3
The main acoustic dimensions on which consonants and vowels differ are spectral and durational whereas tone variation is predominantly based on pitch – f0 height and f0 movement over time (contour) (see [25]). As tones are mainly carried on vocalic elements of a speech signal, it is hypothesized that they are similar to vowels and perceived less categorically. [26], [27]. Given these similarities, musicality could potentially provide a more nuanced understanding of the underlying distinctions between consonants, vowels, and tones.
References:
Abramson, A. (1979). Noncategorical perception of tone categories in Thai. In Journal of the Acoustical Society of America, 61, S66.
Pisoni, D.B. (1975). Auditory short-term memory and vowel perception. In Memory & Cognition, 3, 7-18.
Page 9 sections 4.1 and/or 4.3 the work of Dee Nikjeh is relevant to this discussion (and possibly to considerations in future work) as an effect of type of musical training and music experience influenced performance in production. (Also somewhat relevant in page 3 paragraph 4 discussing differences among musicians.)
Reference:
Nikjeh, D. A., Lister, J. J., & Frisch, S.A. (2009). The relationship between pitch discrimination and vocal production: Comparison of vocal and instrumental musicians. Journal of the Acoustical Society of America, 125, 328-338.
We appreciate your contribution in highlighting and citing the fact that the type of musical experience could be a critical factor in our results, and that the absence of an interaction between various speech categories might be due to the inclusion of both instrumental and vocal musicians in our study. We have included this valuable information in the manuscript. Thank you.
P.3
“Nikjeh et al. [32] showed that instrumental and vocally trained musicians outperformed non-musicians in the production of pitch.”
(…)
“Similarly, it is suggested that the type of musical training and experience influences pitch perception and production abilities, with individual who are vocally trained showing greater accuracy in pitch production [28], [33].”
In the discussion (section 4.1), p 8-9 we have added the following information and critical reflection about the types of musicality that may have influenced consonant, vowel and tone perception as well as production abilities:
“The absence of an interaction between Thai sound contrasts and musicality could be attributed to the fact that pitch perception and production abilities differ depending on the type of musicality, even among musicians, with vocalists exhibiting superior vocal pitch matching skills. In our study, we did not differentiate between musicians with vocal versus instrumental training, although prior research has indicated that vocalists exhibit greater accuracy in pitch perception and production [28], [33]. A more refined categorisation of different musicality types could reveal that musicians with vocal training excel in the domain of lexical tone perception and production, whereas both groups perform similarly for consonants and vowels. However, in our sample, 10 out of 18 musicians had both vocal and instrumental training, while only eight had purely instrumental training, thus precluding a differentiation between the two groups in our study. Future investigations could consider a more precise division of musicians into subgroups (vocalists vs instrumentalist) when investigating speech sound discrimination and production.”
Page 9 section 4.4 on further research would benefit from noting that an additional study of Thai using the back unrounded vowel (which is not a vowel category in English) would provide a better test of whether sound category (vowel versus consonant versus tone) is differentially influenced by musical training.
Indeed, this is a very important aspect for future research. We have added his information to p.10.
“In order to explore the impact of musicality on speech sound categories, future investigations could consider utilizing not only more complex vowels but also vowels that are not part of, or cannot be assimilated to, the native phonemic vowel system.”
Reviewer 3 Report
Dear colleagues,
I think the paper is fascinating.
However, I suggest the following:
1. To change the content of the last passage of the first part "the aim of this study to investigate ..." and describe explicitly the goal and tasks or research questions, and the details regarding the research sample and materials (variables) might be included in the material and methods section.
2. The structure of sections 3 and 4 might be taken into account while specifying earlier the research tasks and (or) questions, now there is no coordination among the data regarding the results and the research tasks, this situation leads to the lack regarding the research logic and paper content transparency.
Some sentences seem to be a bit long with a complicated subordinate structure, though it might be seen as the authors' style
Author Response
Thank you very much for your comment. Please see below the changes (comments and changes in blue) we adapted throughout our manuscript.
- To change the content of the last passage of the first part "the aim of this study to investigate ..." and describe explicitly the goal and tasks or research questions, and the details regarding the research sample and materials (variables) might be included in the material and methods section.
In this section we added the information about specific materials.
p.2-3
The aim of this study is to investigate whether musicality, both formal music training and musical aptitude, influences both the perception and production of non-native speech sounds, specifically Thai consonants, vowels and tones. To this end, 36 English-speaking adults – one group of formally trained musicians and another of non-musicians – are assessed for their musical aptitude and tested for their perception and production of various Thai consonants ([b], [p], and [ph]), vowels([u:], [i:], and [É”:]), and tones (low level, high level, and mid level tone). As both speech and music abilities involve perception and production, we investigate both of these here, first in a perception then in a production task. he same study. Speech perception was evaluated in an identification task that enabled the assessment of not only their ability to discriminate between two speech sounds but also their ability to accurately identify the speech sounds. This task is of particular relevance in evaluating language learning as the ability to correctly identify speech sounds is critical for accurate production of the corresponding speech sounds. The goal of the study is to determine whether, and to what extent, musicality (formal training and/or aptitude) predicts the (a) perception and/or (b) production in (i) consonants, (ii) vowels and/or (iii) tones.
- 2. The structure of sections 3 and 4 might be taken into account while specifying earlier the research tasks and (or) questions, now there is no coordination among the data regarding the results and the research tasks, this situation leads to the lack regarding the research logic and paper content transparency.
We added some structural information to the beginning of the discussion:
p. 8-9
“Although perception and production performance were analysed separately, here they are discussed together in three section in terms of the separate and independent effects of (i) Musicianship, of (ii) Speech Type and how each of these relate to the specificity of any (iii) Cross-Domain Transfer. These three issues are discussed in turn.”
Comments on the Quality of English Language
Some sentences seem to be a bit long with a complicated subordinate structure, though it might be seen as the authors' style